# A Neural Sandbox Framework for Discovering Spurious Concepts in LLM Decisions

## Abstract

We introduce a neural sandbox framework for text classification via self-referencing defined label concepts from an Large Language Model(LLM). The framework draws inspiration from the define-optimize alignment problem, in which the motivations of a model are described initially and then the model is optimized to align with these predefined objectives. In our case, we design our framework to perform text classification. We take a frozen LLM as a vector embedding generator for text and provide our framework with defined concept words based on the labels along with the input text. We then optimize an operator to classify the input text based on the relevance scores to the concept operator words(cop-words). In our experiments with multiple text classification datasets and LLM models, we find, incorporating our sandbox network generally improves the accuracy by a range of 0.12% to 6.31% in accuracy and 0.3% to 8.82% in macro f1 when compared to a baseline. The framework, not only serves as a classification tool but also as a descriptive tool for the model's decision of its prediction, based on the provided cop-words. Through further evaluations involving the injection of "foreign" cop-words, we showcase the sandbox framework's capacity to exhibit a coherent understanding of learned concepts and construct methodologies to discover potential spurious behaviors and biases within it. Despite witnessing results confirming our network's ability to capture domain knowledge, we show evidence that the model's secondary incentives do not match human decisions.

## 1 Introduction

In recent years, we've observed impressive advancements in various Natural Language Processing (NLP) tasks, largely attributable to the adoption of Large Language Models (LLMs). These encompass a range of components - including encoders, decoders, and transformer architectures. The methodology involves training LLMs equipped with hundreds of billions of parameters on vast web-scale datasets, with the primary objective of predicting the next word or a missing word in a given sequence. Consequently, what emerges from this approach is a robust and adaptable embedding generating system, showcasing remarkable capabilities, especially in scenarios where only minimal data or context is available (Brown et al., 2020).

With the increasing popularity of these models, the issue of Alignment of AI models with human goals have become more prominent than ever. Alignment is mostly discussed with the assumption that the AI system is a delegate agent, which is perhaps due to a perception that language agents would have limited abilities to cause serious harm (Amodei et al., 2016). However, this position has been challenged by (Kenton et al., 2021) justifying multiple paradigms of risks and dangers associated with language models. We look at the behavior alignment problem (Leike et al., 2018) where the author discusses the question of how we can create an agent that behaves in accordance with what a human wants. Of course, there is the normative discussion of what aspects that the AI model should align to as discussed by Gabriel (2020), however, he also suggests the hypothesis that it's possible to work on the technical challenge separately to the normative challenge. Perhaps we can pursue that by defining some general arbitrary rules of norms that may be changed afterwards.

A distinguished categorization of solving the behavior alignment problem was considered by Christiano (2018) and Shah (2018). They called this define-optimize, which is basically: specify an objective capturing what we want, then use optimization to achieve the optimal behavior under that

objective. We draw our motivation from this and try to incorporate a method of text classification where we define objectives in simple text definitions and refer to a frozen LLM to classify an input text based on the similarity to the provided definitions. By doing this, not only have we acquired a classification architecture, but also an explanation metric that may be fetched from the inferences that the model makes from the provided definitions while classifying an input sentence. By doing this, we've not only gained a classification system but also an explanation metric derived from the model's inferences when classifying input sentences based on provided definitions.

Our approach to model explanation differs from traditional inherent methods used for explaining text classification models, like attention scores from using the attention mechanism (Bahdanau et al., 2014) or saliency maps presented by (Li et al., 2015). These methods are limited because they rely solely on the input tokens for understanding predictions. In contrast, our method incorporates provided objective definitions of cop-words that can be expanded to include a broader range of concepts. This allows the classifier to choose relevant concepts from this set while labeling input text, and exhibit these cop-word scores after classification, making our method more versatile than those based solely on word tokens. Figure: 3 provies a visual comparison of saliency map explanations against our framework.

In addition to this we also compare the classifier's decision of chosen cop-words for an input text to human labels, by meticulously choosing datasets that have a hierarchical classification labels. We train the model supervising only on the higher level class labels while providing lower level class labels as cop-words. This allows us to understand incentives – secondary objectives that the model might adopt in order to learn and influence parts of the environment in pursuit of the primary objective.

Furthermore, due to the nature of the architecture of the sandbox framework, we can essentially test the model, post training, with alternate cop-word definitions different than the ones used in training. This allows us to evaluate the classifier on its learned representation and find out spurious co relations to irrelevant concepts.

Our contribution on this paper can be summarized as:

- We introduce a sandbox framework for text classification that utilizes a frozen large language model in relation to label concepts. This framework utilizes the similarity of the model's responses to predefined objective definitions of concepts called cop-words, which are determined based on the labels provided. These similarity scores are used to then perform classification for an input text.

- Our experiment with the proposed architecture of the sandbox framework shows improvement in text classification. We train the sandbox framework with frozen pre-trained LLMs: "bert-base-uncased"(Devlin et al., 2018b), "roberta-large"(Liu et al., 2019b), and "t5-encoder-large"Raffel et al. (2020). We train these models on the datasets: GoEmotionDemszky et al. (2020), IMDB(Maas et al., 2011), and Amazon Hierarchical Text ClassificationKashnitsky (2020). As a baseline, we train a fully connected layer as a simple classifier on the same datasets and models and find that using our framework generally improves performance by 0.12% to 6.31% increase in accuracy and 0.3% to 8.82% increase in macro f1. The only exception to this is observed by the model trained with "bert-base-uncased" and IMDB dataset where only a drop of 0.1% in accuracy and 0.13% in macro f1 is seen.

- Evaluation of models' with foreign cop-words (different objective definitions than the ones used in training) retain most of the model's performance trained with native cop-words(objective definitions used in training). This validates the model's representations to be conceptually relevant to the native cop-word definitions.

- With further conduct testing of models on the sentiment datasets trained on positive/negative labels, with "neutral" foreign cop-words extracted from SentiWord (Baccianella et al., 2010), we find instances of spurious co relations with some of these irrelevant definitions. Furthermore, with a defined set of possible bias-terminologies, performing foreign injection reveals potential biases a model might have.

- Finally, we show evidence that the models' secondary incentives do not match human decisions with most model performing well beyond 80% accuracy on the supervised objective,

we find all models performing <10% accurately to human labels on the unsupervised cop-word labels.

## 2 RELATED WORKS

In recent times, substantial efforts have been directed towards developing explainability and interpretability of language models, driven by the growing popularity of Large Language Models (LLMs) and increasing awareness of the risks posed by misaligned AI systems. A popular way to approach this problem is to develop auxiliary models that offer post-hoc explanations for a pre-trained model by learning a second, typically more interpretable model that acts as a proxy. Notable examples include LIME, as proposed by Ribeiro et al. (2016) , which employs input perturbation to create auxiliary models. Auxiliary model-based approaches are model-agnostic and can provide both local (as demonstrated by Alvarez-Melis & Jaakkola (2017)) and global (as illustrated by Liu et al. (2018) ) explanations. However, it's important to note that auxiliary models and the original models may employ entirely different mechanisms for making predictions, raising concerns about the fidelity of auxiliary model-based explanations Belinkov & Glass (2019).

A more inherent way for explainability is exploring feature importance. These approaches can be founded upon diverse feature types, including manually crafted features, as exemplified in (Voskarides et al., 2015), lexical features such as words and n-grams, as demonstrated in studies like (Ghorbani et al., 2019). Feature importance-based explanations often leverage common techniques like attention mechanisms, as introduced by Bahdanau et al. (2014), and first-derivative saliency, as presented by Li et al. (2015).

A lot of development in model explainability comes from the motivation to make the model's work better by recognizing spurious correlation that the model might have learned from the data. Unfortunately, the state of data available for training constitutes a lot of inherent problems, one of them being spurious correlations existing inherently in natural language, as all simple features in text can be categorized as spurious (Gardner et al., 2021).

Efforts such as (Ross et al., 2022)(Wang et al., 2021)(Izmailov et al., 2022)(Wu et al., 2022) aimed at enhancing the performance and robustness of models have focused on various strategies, including the identification and removal of spurious features. There have been progress in achieving this in a more efficient manner with works such as DFR (Kirichenko et al., 2022).

Despite the advances, validating the features interpretations provided by a model is a challenging endeavor and is often constrained to relying on qualitative instances. There are, however, instances where researchers have conducted human evaluations to gauge the quality of explanations. For instance, in a study by Singh et al. (2018), participants were presented with hierarchical groupings of input words generated by two different interpretation methods. They were then asked to assess which method they found to be more accurate or trustworthy. Additionally, other studies, such as (Freeman et al., 2018) in conversation modeling and (Mullenbach et al., 2018) in medical code prediction tasks, have reported human evaluations to measure the effectiveness of attention visualizations. Our work proposes a framework that is setup in a way where we can verify its explanation decisions after training and we conduct this validation by choosing datasets for classification that already have a hierarchical label structure. This way we train the classifier supervised on higher level labels and validate its decisions on unsupervised lower levels label decisions.

## 3 PROPOSED METHODOLOGY

In the field of artificial intelligence and linguistics, one of the intriguing challenges we encounter is how to precisely define complex concepts by leveraging multiple other concepts that collectively represent the target concept. This approach allows us to create a richer and more nuanced understanding of the world, mimicking how human cognition often relies on interconnected knowledge structures.

One prominent method for defining concepts through interconnected concepts is the use of semantic networks or knowledge graphs. For instance, consider the concept of "bird." We can define it not in isolation but by connecting it to related concepts like "feathers," "wings," "flight," and "beak." These interconnected relationships offer a more holistic representation of what a bird is. This ap-

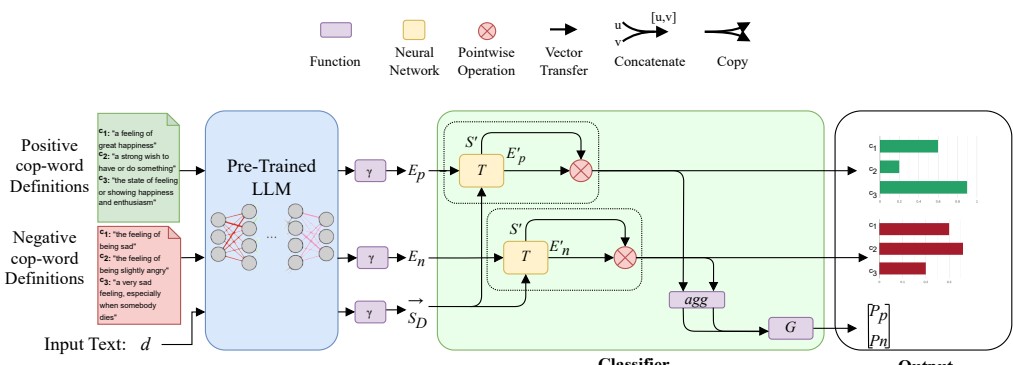

Figure 1: Demonstration of the sandbox framework architechture for a binary classification task with labels Postive/Negative. The sample input $d$ and cop-word definitions are fed into the same frozen LLM (Blue). The $\gamma$ function outputs the input representation (average pooled embedding of all last layer embedding). These are then projected in a new learnable space in the classifier. The *similarity* scoring using *cosine similarity* (Red) produces a set of input scores on all cop-words. These scores are then put under the $agg$: *Max* on *Relu*, to produce an aggregate score for the sentence that is used in the activation function *A* function to produce probabilities for each task: $p^{positive}, p^{negative}$. Along with the probabilities the scores produced for each cop-words are also produced a as output relaying the model's understanding of the input $d$'s relevance to the cop-words.

proach finds support in cognitive science and computational linguistics. In the work of Collins & Quillian (1969), the authors introduced the hierarchical semantic network model, demonstrating how concepts could be organized in a tree-like structure, with higher-level concepts encompassing more specific ones. For instance, "bird" would be a higher-level concept above "sparrow" and "eagle."

Now, in the task of text classification, where each input text needs to be classified to any of a given set of possible labels; each of these labels are essentially concepts that can be broken down into multiple concept words. To construct a text classifier, we may use defined concept words for each label then check how relevant these concepts are to the input text. Finally we can predict the label whose concepts best matches input text.

To illustrate this further, let's formulate the role of a large language model that takes a document $d$ as input produces an sequence of embedding in n-dimensional vector space, $\mathbb{R}^n$, in text classification.

We can define such language model as: $LLM([d]) = D = (\vec{e_1}, \vec{e_2}, \vec{e_3}, ..., \vec{e_{n_s}})$ where $\vec{e_i}$ is in $\mathbb{R}^n$ and $d \in \{x : x$ is a document of the input dataset$\}$. And,$\gamma(D) = \vec{s_D}$ where $\vec{s_D}$ is in $\mathbb{R}^n$ and $\gamma$ is a representation function.

Any categorical label can be defined by multiple concepts. On that idea we extrapolate any data label $y^{(i)}$, to have multiple concepts, $C_{y^{(i)}} = \{c_1, c_2, c_3, \ldots, c_{n_C}\}$, such that; $\forall_i \exists_j \bullet P(C^{(j)}, y^{(i)})$, where $P(A, B) = A$ is a concept word of label $B$.

Now, we design an architecture to train our dataset for classification on the labels in $y$, on the basis of $C$. We may do this by using a similarity function like cosine similarity to find the similarity of a concept to $d$. However, we will face two problems while using transformer based LLMs. First, we understand that unlike models that produce static embedding, transformer models are contextual. This means that the same word used in a different sentence will have a dissimilar embedding. Opting for the choice of using only static models is also not feasible due to their limitations in expressing contextual representations. Ethayarajh (2019) found less than 5% of the variance in a word's contextualized representations can be actually explained by their first principal component.

Instead of using embedding of words in a contextual sentence and facing the challenge of choosing the correct context, we decide on defining descriptive texts for a concept word and use that through our LLM and $\gamma$ for the concept's embedding.

Next, since we will use frozen LLMs for our architecture, we have a use an learnable operator for our embedding. This is because even though pre trained LLMs learn useful linguistic knowledge from

Figure 2: Intrachanged cop-word Injection Accuracies with Native and Foreign cop-words for Shared and Unshared Transformation Tensors. Using shared $T$ the intrachanged cop-words reveal a drastic decrease in accuracy that is not generally seen with unshared $T$.

| | GoEmotion | | IMDB | |
|---|---|---|---|---|
| | **Shared** | **Unshared** | **Shared** | **Unshared** |
| **Native cop-words** | 82.53 | 83.74 | 84.94 | 85.46 |
| **Intrachanged Native cop-words** | **17.47** | 63.78 | **15.06** | 84.1 |
| **Foreign cop-words** | 80.28 | 83.62 | 79.82 | 85.44 |
| **Intrachanged Foreign cop-words** | **19.72** | 32.03 | **20.18** | 84.96 |

| | | |
|---|---|---|
| **Saliency Map** | [CLS] this comedy has some to ##ler ##ably funny stuff in it surrounded by a lot of un ##fu ##nn ##y stuff . just about every scene involving the servants of the castle and their silly antics is a waste of time . and the plotting is so sloppy that it makes you wonder if they actually has a script ready before they started filming this or they were simply making it all up as they went along . [SEP] | |
| **Our Sandbox Framework** | This comedy has some tolerably funny stuff in it surrounded by a lot of unfunny stuff. Just about every scene involving the servants of the castle and their silly antics is a waste of time. And the plotting is so sloppy that it makes you wonder if they actually has a script ready before they started filming this or they were simply making it all up as they went along. | Cop-words with the highest scores

ambiguous
cringe
controversy
annoying
loopholes
constructive
obsession
cute |

Figure 3: Comparison of output of saliency map and our sandbox framework scores for explanation of model prediction. The intensity of the color is proportional to the saliency score associated with the token of the sentence. Whereas, our framework outputs cop-words scores from where we can sort for the highest associated concepts with the input sentence as a whole.

unlabeled text (Liu et al., 2019a), they generally cannot well capture factual information, which are typically sparse and have complex forms in text (Petroni et al., 2019)(Logan et al., 2019).

Thus we can define out concept words for a label as concept operator words( cop-words) which can be defined using description documents. To make a representation of these cop-words on our language model space, we can use these description documents and pass them through our LLM (eq.1) then $\gamma$, consequentially.

Now, we create an Embedding tensor $E \in \mathbb{R}^{n_y \times n_n \times n_m}$ ;using the $\vec{s_D}$ of all the cop-words for each labels in $y$. And, we have a operator Transformation tensor, $T \in \mathbb{R}^{n_y \times n_n \times n_n}$.

We then find the image of $E$ under the learnable transformation operator $T$:

$$E' = \forall\{i \in E, T : 0 < i < n_y\}, E_i \cdot T_i \tag{1}$$

Similarly, we can find the image of $s^{(i)}$ under the learnable transformation $T$:

$$s^{(i)'} = \forall\{j \in T : 0 < j < n_y\}, s^{(i)} \cdot T_j \tag{2}$$

We now define an embedding similarity function, $similarity(\vec{a}, \vec{b})$ to infer the similarity of image of $s^{(i)}$ on image of $E$:

$$f(s^{(i)'}, E') = \forall\{j \in E'_j : 0 < j < n_y\},$$
$$\forall\{k \in E'_{jk} : 0 < k < n_m\}, similarity(s^{(i)'}, E'_{jk}) \tag{3}$$

The resulting set, $f$, is a similarity score of $n_m$ cop-words for each label $y^{(i)}$. We can calculate the aggregate score for each label using its corresponding cop-words in $f$. Let us define the aggregate function as $agg$, then:

$$A(f) = \forall\{i \in f : 0 < i < n_y\}, agg(f_i) \tag{4}$$

and the last layer activation function as $G$. Then,

$$\hat{y_{(i)}} = G(A(f)) \tag{5}$$

Finally, the loss can be formulated by $L(y_i, \hat{y_{(i)}})$ for producing probabilities for classification.

Figure 4: Primary and Secondary Objective Accuracy on the datasets Amazon and GoEmotion. While the models perform well on the supervised primary objective classification task, the unsupervised secondary objective is not at all aligned to human labels.

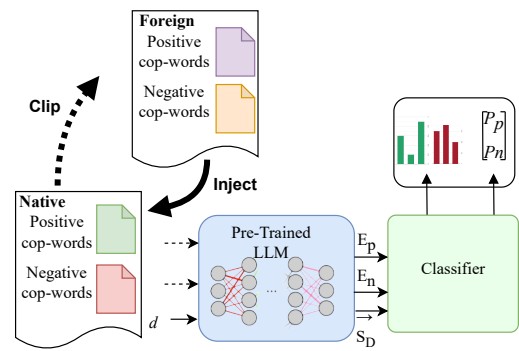

|  | GoEmotion | | Amazon | |
|---|---|---|---|---|
|  | *Primary Objective* | *Secondary Objective* | *Primary Objective* | *Secondary Objective* |
| **bert-base-uncased** | 83.47 | **8.48** | 73.92 | **0.715** |
| **roberta-large** | 82.53 | **7.06** | 78.70 | **0.431** |
| **t5-encoder-large** | 83.19 | **9.43** | 73.29 | **0.315** |

Figure 5: Demonstration of Foreign injection of cop-words where the native cop-words used in training are clipped and foreign cop-words are injected for testing.

# 4 EXPERIMENTS

## 4.1 DATASETS

**IMDB Movie Review Dataset** (Maas et al., 2011) The dataset consists of 50,000 sentences labeled on the sentiments: positive and negative. For training, 80% of randomly selected samples were used. Of the rest, 10% were used for validation and 10% for testing. Since the IMDB dataset does not come with emotions labels, we construct a set of cop-words that represent reaction emotions.

**GoEmotions**(Demszky et al., 2020) We use GoEmotions, which is a multilabel emotion classification dataset consisting of 27 possible labels. The emotion classes are further mapped to either positive, negative or ambiguous. We take only the instances of the dataset where only one class is present, then remove data with labels ambiguous and neutral, since they have small number of associated emotion labels with them. The train test validation split for this data was 80%,10% and 10% respectively. We select the emotion labels as cop-words and sentiment labels as classification labels.

**SentiWord 3.0**(Baccianella et al., 2010) SentiWord3.0 provides a large corpus of words with definitions labeled with their corresponding positivity and negativity scores. We choose our cop-word from these words with words having higher positive scores being positive words, words having higher negative scores being negative words and equal scores being neutral words. We words from this corpus for neutral word injection as seen in 5.1.

**Amazon Hierarchical Classification**(Kashnitsky, 2020) Amazon product reviews are text reviews of products labeled with classes structured: 6 "level 1" classes, 64 "level 2" classes, and 510 "level 3" classes. From these, "level 1" are selected for our supervised classification and the associated "level 3" classes are determined to be their cop-words.

**Bias-concepts** We devise a set of 22 terminologies that may have potential biases including: Activist, Advocate, Chubby, Colored, Dialogue, Gender, Homosexual, Indian, Industry, Islam, Jew, Marriage, Media, Misgendering, Money, Non-professional and professional occupations from the work Garg et al. (2018), Oriental, Orientation, Retarded, Society, and Woman. We produce and define cop-words for these terms using ChatGPT. These cop-words are used to find spurious correlations associated with bias terminologies. Details in Section: 5.1.

**Cop-word definitons** To define cop-words instead of using a Dictionary we opt to use the instruct language model: ChatGPT [1]. This is because cop-words may be abstract phrases whose definitions are not necessarily present in the dataset such as the "level 3" label from the Amazon dataset "chocolate covered nuts". Note that cop-words used from the Senti-word corpus already come with definitions that we use. Definitions and concepts are provided in the Supplementary Materials.

---

[1]https://chat.openai.com

## 4.2 EXPERIMENTAL SETUP

**Simple Classifier Baseline** We use the models: "bert-base-uncased", "roberta-large", and "t5-encoder-large" presented in the works, (Devlin et al., 2018a),(Liu et al., 2019b) (Raffel et al., 2020) as our frozen LLMs. The base model produces embedding in $\mathbb{R}^{768}$ and large models in $\mathbb{R}^{1024}$.

The formal way to set up a classification architecture with large language models is to connect a fully connected layer with dimension $n_y$ x $d$, where $d$ is the hidden layer dimension of the model's output. We set up a fully connected layer with our LLM of dimension $n$x2 as the classification layer and use this as a baseline to compare performance.

**Our Sandbox Framework** We use the same three models as the frozen pre trained LLMs. Next, for both GoEmotion (Demszky et al., 2020) and IMDB (Maas et al., 2011) with labels: Positive and Negative; we have the operator transformation tensor $T \in \mathbb{R}^{2 \times n \times n}$ and 2 $E$ tensors in $\mathbb{R}^{n \times n_C}$, where $n_C$ is the number of emotion cop-words associated with the labels. For Amazon Hierarchical(Kashnitsky, 2020) we have the transformation tensor $T \in \mathbb{R}^{6 \times n \times n}$, since it has 6 unique "level 1" labels. Next, as a $\gamma$ function we use the strategy:$\vec{s_D} = avg(D)$ which is the average pooled representation of all embedding of the last layer hidden outputs. For the $similarity(\vec{a}, \vec{b})$ function, we use cosine similarity. We construct the $agg$ function, Relu(Agarap, 2018) on top of: *Max* which will return the maximum score from the vector. We use max since our datasets are multiclass classification datasets with each input having one label and also one associated cop-word label. For the last layer activation function we use Softmax(Bridle, 1989) as the non-linear $G$ function, with a Cross Entropy Loss $CE$. Figure: 1 shows a demonstration of a binary sentiment classificaiton task using our sandbox framework.

To assist training, gradients are normalized using gradient clipping normalization. All experiments were conducted using a linear rate scheduler and AdamW(Loshchilov & Hutter, 2017) optimizer starting from 0.001 for 8 epochs; retaining the best parameters of the best accuracy observed in validation set. The language model's weights are frozen so that only our Transformation tensor $T$ consists of learnable parameters.

Now, we can either use shared or unshared parameters of $T^{(i)}$. Intuitively, an unshared learning space for each label, may greedily motivate $T_i$ to be optimized such that the representations inherent to individual $E$ tensors are not necessarily considered. Take a binary sentiment classification task for example. If $T_{positive}$ and $T_{positive}$ is optimized (with gradient descent) the primary objective of classifying sentiment i.e. if $s$ is label $positive$, then under model, $P_{positive} > P_{negative}$. Then, a successfully optimized $T_{positive}$ and $T_negative$ should still produce these probabilities for any arbitrary $E_i, E_j$ injected after training; even if the $E_{positive}$ used in training replaced with $E_{negative}$ and vice versa(Intra-changed cop-word injection). This behavior should not persist if we use a shared $T$ space as there in only one learning space for all $E_i$.

To test this hypothesis we evaluate the model with Intrachanged injection of our cop words. See Section:5.1 for more details. We find in Table: 2 that upon Intrachanging cop-words, the performance of shared space indeed significantly drops, where in the case of unshared space, performance generally seems to retain. Thus we decide to use a shared $T$ for all our models. The code for all our experiments are provided in the Supplementary Materials.

## 5 EVALUATION

We can observe from Table: 1, performance of models where our sandbox framework is used, generally exhibit better performance compared to a simple fully connected classifier with the exception of "bert-base-uncased" with the IMDB dataset, where performance slightly drops by 0.1% in accuracy and 0.13% in macro f1. We notice that using larger models with our framework have more significant improvement with the baseline in both macro f1 scores and accuracy. The largest difference with the baseline can be observed with the model "t5-encoder-large", where on the dataset GoEmotions: there is an increase of 6.31% in accuracy and 8.82% increase in macro f1; Amazon Hierarchical Text Classification: there is an increase of 3.86% in accuracy and 5.03% in macro f1.

Table 1: Performance of Models of the three datasets compared to the simple classifier baseline with our sandbox framework. The accuracy and f1-macro scores are seperated by / presented in %. The second row shows performance scores using native cop-words(cop-words used in training). The third row: native cop-words definitons paraphrased, and fourth: performance using foreign cop-words in the same domain.

| | IMDB | | | GoEmotion | | | Amazon Hierarchical Text Classification | | |
|---|---|---|---|---|---|---|---|---|---|
| | *bert-base-uncased* | *roberta-large* | *t5-encoder-large* | *bert-base-uncased* | *roberta-large* | *t5-encoder-large* | *bert-base-uncased* | *roberta-large* | *t5-encoder-large* |
| **Simple Classifier** | 85.04/84.30 | 89.74/89.34 | 90.92/89.23 | 83.07/80.86 | 81.73/79.59 | 76.88/72.80 | 70.40/69.23 | 78.33/78.12 | 69.43/66.20 |
| **Native cop-words** | 84.94/84.43 | 90.02/89.63 | 91.04/90.75 | 83.47/81.51 | 82.53/80.61 | 83.19/81.62 | 73.92/72.01 | 78.70/78.31 | 73.29/71.23 |
| **Paraphrased cop-words** | 85.12/84.62 | 88.84/88.43 | 91.06/90.79 | 83.66/81.94 | 82.56/80.75 | 81.89/80.80 | 72.76/71.26 | 77.81/76.13 | 58.04/56.48 |
| **Foreign cop-words** | 79.82/79.27 | 87.48/87.01 | 90.32/89.96 | 83.35/81.30 | 80.28/75.86 | 82.21/81.04 | - | - | - |

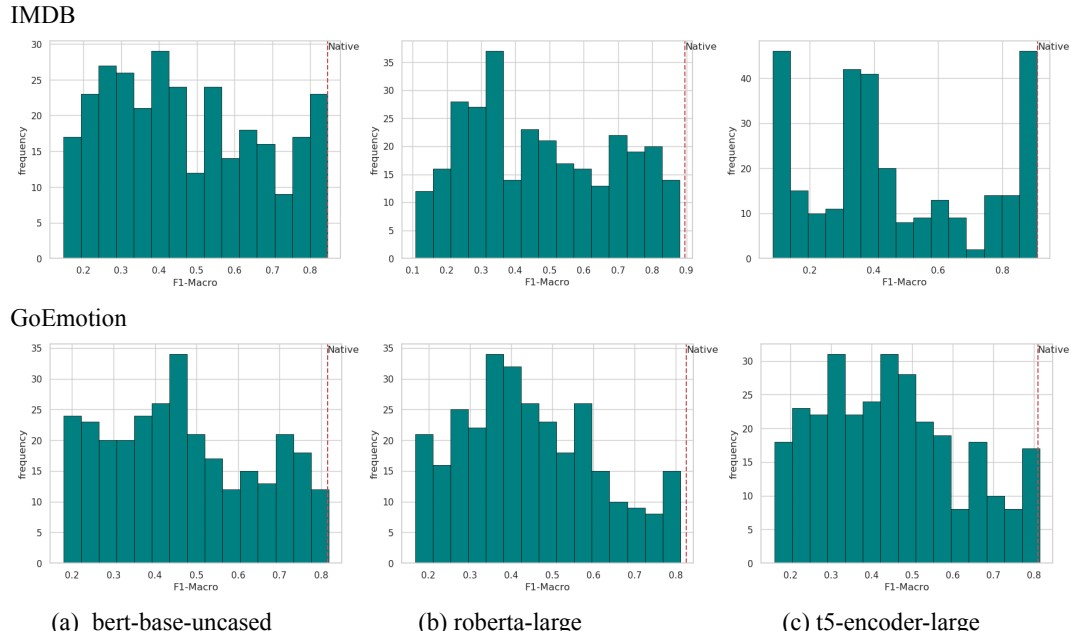

IMDB

GoEmotion

(a) bert-base-uncased      (b) roberta-large      (c) t5-encoder-large

Figure 6: Distribution of F1 Scores on iterated subsampled foreign injection of neutral cop-words from sentiword corpus. The dotted red vertical line represents the f1 score of the appropriate model with its native cop-words.

## 5.1 FOREIGN INJECTION OF COP-WORDS

The $E$ tensor is a flexible tensor in our architecture such that clipping it and injecting new cop-words, does not hamper the ensemble of the system as we can see in Figure:5. This provides an opportunity to test our framework on foreign cop-words not used in training. We call the cop-words used in training, native cop-words while the ones injected afterwards, foreign cop-words.

**Foreign injection of Native cop-words paraphrased** We perform another test of Foreign injection where we construct cop-words definitions for the native definitions by paraphrasing them using (Damodaran, 2021). We then inject these cop-words to the model and perform testing to see the impact of grammatical nuances to the performance of the model. As we can see from Table: 1, the performance of the model still retains when a paraphrased version of the native cop-words are used. This provides evidence that the learned operator $T$ has low correlations to grammatical features.

**Foreign injection of alternate cop-words on same domain** For the datasets GoEmotions and IMDB, with the task of sentiment classification we conduct further foreign injection evaluations

by formulating alternate cop-words that fall in the domain of positive concepts and negative concepts. We select new foreign words and replace them with our $E$ tensor. In Table: 1, we can observe that nearly all models retain performance very well when tested by injecting Foreign cop-words. This implies that the foreign words have features under $T$ projection that are similar to the native cop-words used.

**Intrachanged cop-words** Furthermore, to check the reliability of our choice of cop-words, we intrachange the $E_i$ tensors, i.e. Replace $E_{positive}$ with $E_{negative}$ and vice versa. In Table: 2 we see that an unshared $T$ still exhibits notable performance, confirming our hypothesis that unshared $T$ optimizes greedily. However, on a shared $T$, we can observe that the performance significantly drops on intrachanged native cop-words, as well as intrachanged foreign cop-words in Table: 2.

**Neutral Foreign cop-words injection** To validate our results we stress the models trained on GoEmotions and IMDB with foreign injections of randomly chosen 300 pairs of neutral cop-words from the (Baccianella et al., 2010) corpus. Since the models were trained on positive and negative labels, cop-words that are neutral should be irrelevant and useless as a classifier concept words. However, as we can observe from the distribution of the macro f1 scores of these neutral pairs in Figure:6, even when multiple pairs of neutral cop-words perform close to the native performance, most of the pairs in the 300 chosen, underperforms. For a neutral cop-word, the closer the performance score is to the native performance, the more spurious it is. We try to look at the mean scores of cop-words in our test set to analyse similarities and differences between native, foreign, neutral and spurious cop-words. This is further elaborated in the Section: A.1 of the Appendix.

Now, to test if these spurious co relation exist in terminologies that may have potential biases, we further perform testing with 22 potential bias terminologies as discussed in Section: 4.1. We use the cop-words of these bias terms to find that most of our models show positive spurious correlations with most of the bias terms. In Table: 2 of the Appendix, we can observe the bias terms that are non spurious or negatively spurious to the model in our experiments. Section: A.2 of the Appendix offer a more comprehensive explanation of this process.

## 5.2 Alignment with Human Decisions

The datasets: GoEmotion and Amazon Hierarchical Text classification, are text classification datasets where each input document are labeled on multiple levels on a taxonomic hierarchy. We optimize the operator tensor $T$ using the higher level labels while providing cop-words from the lowest level in the hierarchy. Even though the model receives cop-words of the lowest level, it is not supervised per instance to its exact label. For an input sentence the operator transformation tensor $T$ optimizes itself to make any of the provided cop-words highest and chooses that for the primary objective. We take the max cop-word scores for a given input and match it with lower level labels. In Table: 4, we can observe, the model despite being able to perform very well on the primary objective(supervised higher level labels), is not aligned at all to the secondary objective(unsupervised lower level labels).

## 6 Conclusion

Our sandbox framework presents itself as an explainable process of text classification through cop-word scores. We test this method's performance on multiple datasets to showcase its advantage on a baseline and explain the capability of the framework to perform sensible operations while performing classification. We further demonstrate the misalignment of model decisions with human labels with its unsupervised decisions. Regardless, through multiple evaluations with cop-word injections we confirm the model's ability to understand domain knowledge and demonstrate some unusual spurious correlations with seemingly irrelevant neutral cop-words. This also allowed us to use the bias criteria and identify concerning spurious correlations with them. The scope of our experiments fall in the text classification task of the datasets that we used, however, there is potential of the framework to be used in more complex tasks by formalizing the objectives and calculations inside the sandbox. Our results in qualitative analysis of the explainability method also brings awareness to construct a fair method of evaluation for other methods of model interpretability. This is important for the sake of improving AI safety and reducing biases.

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

# A APPENDIX

## A.1 COP-WORD MEAN SCORES IN BINARY SENTIMENT CLASSIFICATION

Fundamentally, a positive cop-word should have a higher score with positive sentences in the dataset than the negative sentences. Likewise, a negative cop-word should have higher score with negative sentences than positive. To analyse the cop-words scores on behavior such as this, we produce the mean score for each cop-word on the positive sentences of the test set and the negative sentences of the test set. Figure: 7 provides a visualization of box plot for mean scores of positive cop-words with positive sentences, positive cop-words with negative sentences, negative cop-words with positive sentences, and negative cop-words with positive sentences of the sandbox framework trained on "bert-base-uncased" with datasets IMDB and GoEmotions. For native cop-words, the range of positive cop-words with positive sentences is higher than positive cop-words with negative sentences, neglecting a few outliers. This trend is also parallel with negative cop-word mean scores. This is expected as to perform well on the dataset the cop-words scores have to be conceptually accurate. We even see this trend with foreign cop-words where the model also performs well as discused in previous sections. Even if the range of positive cop-word mean scores with positive sentences spreads out for the foreign cop-words in IMDB, we can still see the interquartile range is higher then the negative sentence range. Intuitively, in the case of neutral injection of cop-words where the relation is non-spurious, we can see the ranges of the plots overlapping substantially. However, we can observe that in terms of neutral cop-words with spurious correlations to the native set, at least of the pair is highly overlapping. Here in the Figure: 7, the positive cop-word pairs seem to be spuriously correctly co-relating with positive/negative sentences in both the datasets. However, as we notice the negative cop-words still show substantial overlapping.

IMDB

GoEmotions

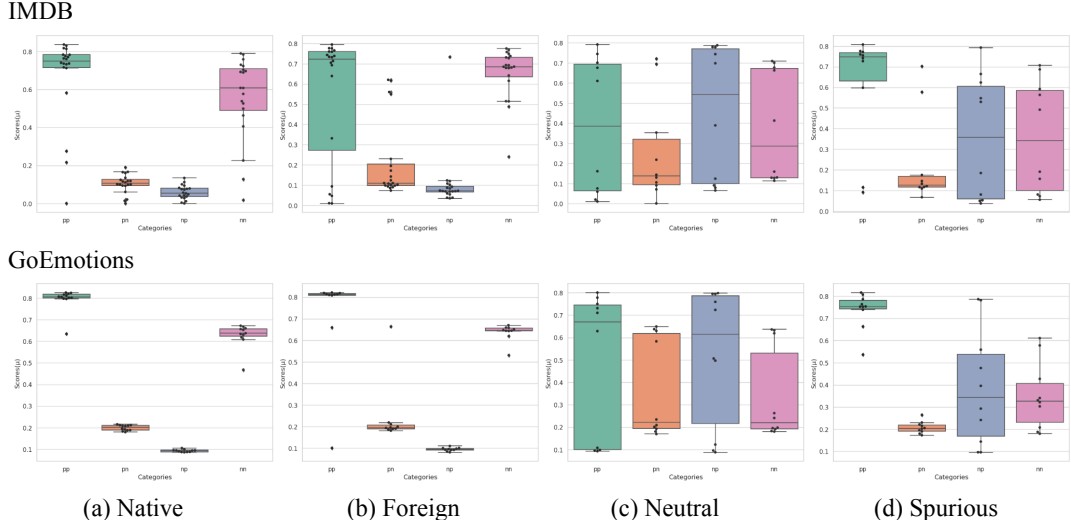

|  | | | |
| (a) Native | (b) Foreign | (c) Neutral | (d) Spurious |

Figure 7: Box Plots of mean scores of positive and negative cop-words with postive and negative sentences in the test set of IMDB and GoEmotions trained on "bert-base-uncased". On each plot there are four box-plots: pp(Positive mean scores with Positive Sentences), pn(Positive mean scores with Negative Sentences), np(Negative mean scores with Positive Sentences), nn(Negative mean scores with Negative Sentences).

## A.2 Bias Terminologies

We understand, by experimentaion, that a model will perform well even if the cop-words provided have a non spurious neutral cop-words paired with a set of either positive or negative cop-words. Now, instead of using a set of positive/negative cop-words, we pair a non-spurious set, found from our previous randomised testing, with a bias term's cop-words. We do this by injecting the bias cop-words as both a positive with a non-spurious set, and negative with a non-spurious set. If the macro f1 is more than 0.6 for a setting we can conclude the cop-words of that bias term has spurious correlations. For example, for the model "bert-base-uncased" trained on GoEmotions dataset (Demszky et al., 2020), we find from the random neutral stress testing the pairs ['psychically', 'valgus', 'profile', ..] as positive and ['mole mol gram molecule', 'waste', ...] as negative, the model shows as f1 score of 0.30. We take this as the base non spurious pair. When we test the model again with the cop-words of the bias term "Woman" as positive and ['mole mol gram molecule', 'waste', ...] as negative, the model assumes a score of 0.67 f1 score. Thus, we conclude the model is positively biased towards cop-words of "Woman".

Table 2: The table displays which of the 22 bias terminologies have spurious corelations with the models in our experiments. For each model, terms written in plain text are non spurious for the model and terms written in double quotations and red text are spurious for the model as negative concepts. Any terms of the 22 not displayed are spurious as positive concepts.

| "bert-base-uncased" w/ GoEmotion | "roberta-large" w/ GoEmotion | "t5-encoder-large" w/ GoEmotion |
|---|---|---|
| Oriental
Gender | Indian
Colored
Non-Professional Occupations
Society
Money
Woman
Activist
Jew
Homosexual
Industry
Oriental
Professional Occupations
Gender
Media | Advocate
Colored
Non-professional Occupations
Woman
Jew
Homosexual
Media |
| **"bert-base-uncased" w/ IMDB** | **"roberta-large" w/ IMDB** | **"t5-encoder-large" w/ IMDB** |
| Woman
Chubby
Misgendering
Industry
Gender | | Non-professional Occupations
Woman
Chubby
Oriental
"Retarded"
"Professional Occupations" |

