# OpenReview forum: "A Neural Sandbox Framework for Discovering Spurious Concpets in LLM Decisions"
_ICLR.cc/2024/Conference — Submitted to ICLR 2024_

### Official Review · Reviewer_cfUj · 2023-10-30

**Soundness:** 1 poor
**Presentation:** 1 poor
**Contribution:** 2 fair
**Rating:** 1
**Confidence:** 4

**Summary:**

This paper proposes a “sandbox” framework for text classification. In this framework, the LLM is kept frozen. The input texts contain both the problem input and the textual descriptions of labels (”concept operator words”, or cop-words). This frozen LLM computes the cosine similarities between the input texts and the cop-words. These similarity scores are then passed into an aggregate function, leading to the prediction result.

With experiments on multiple text classification datasets and LLM models, this paper finds that this sandbox network generally improves the accuracy, compared to a baseline (direct text classification without cop-words).

This paper shows evidence that the model’s secondary incentives do not match human decisions.

**Strengths:**

- The explanation and alignment of models is an important problem, and the idea to do model explanation and/or alignment by routing the models through text descriptions of concepts is an interesting one.
- Through experiments, this paper shows the proposed approaches have better performances than baselines.
- There are some ablation studies showing the effects of different configurations of cop-words.

**Weaknesses:**

- How the proposed method is relevant to behavioral alignment is unclear. As far as I understand, model alignment requires adapting/training the models in some ways, but the proposed framework in this paper keeps the LLM frozen.
- The experiments are not strong enough to support the goal indicated in the title of this paper, i.e., “discovering spurious concepts”. It is unclear to me how the experiments are relevant to *spurious* concepts (the cop-words only describe the concepts).
- The experiments are also not strong enough to reveal the additional benefits in model explainability. I was expecting some quantitative or qualitative results comparing the “model’s secondary incentives” and “human decisions”, but these are not clear from the experiments.
- Crucial components in the framework are not described. How the cop-words are determined is not elaborated (I can see some sporadic descriptions, e.g., “defined using description documents”, but what documents are these description documents?)
    - Relatedly, this paper spends the space lavishly describing some details that are not central to the methodological contributions, e.g., how the components of the frameworks can be written in equations.
- There are multiple typographical errors throughout this paper. Many sentences can’t be parsed. Several citations have incorrect formats (e.g., citep vs citet). I think the readability can significantly benefit from a round of careful proofreading.

**Questions:**

Could you elaborate the procedure to define the cop-words?

---

### Official Review · Reviewer_1xDE · 2023-10-31

**Soundness:** 3 good
**Presentation:** 3 good
**Contribution:** 3 good
**Rating:** 3
**Confidence:** 3

**Summary:**

The paper introduces a neural sandbox framework designed for text classification. The framework is supplied with specific concept words (cop-words) related to the classification labels, alongside the input text. The system then optimizes an operator that classifies the text based on the relevance of these concept words with the input text.

The paper highlights the framework's potential as an evaluative tool to uncover any biases or spurious behaviors. This is achieved by introducing "foreign" or unrelated concept words and observing the model's reactions, allowing for an understanding of the model's underlying learned concepts and potential biases. Comparing the model’s chosen concepts for a specific input text with human-selected lower-level labels, it also offers insight into the model’s decision-making based on the provided concept words.

In addition, the paper demonstrates that introducing this neural sandbox network improved text classification accuracy and macro f1 scores across multiple datasets when compared to a standard baseline.

**Strengths:**

1.The evaluation method involving "foreign" cop-words is an insightful approach to understanding and identifying spurious correlations, biases, or other undesired behaviors in the model.

2.The framework’s ability to not only classify text but also provide insights into the model's decision-making process using cop-words is a notable advantage. This could bridge the gap between model interpretability and performance.

**Weaknesses:**

1.The framework's overview can be described as a retrieval methodology. In this method, databases comprise concept words or descriptions that match high-level labels. Models then infer by measuring the similarity between the input text and these database entries. From this point, the proposed method seems to have limited novelty.

2.The presentation of mathematical formulas lacks clarity, making it challenging to understand and replicate the research. I recommend revising the mathematical sections with clearer definitions and examples for better comprehension and accessibility.

**Questions:**

I have no questions.

---

### Official Review · Reviewer_57Af · 2023-11-01

**Soundness:** 1 poor
**Presentation:** 1 poor
**Contribution:** 1 poor
**Rating:** 1
**Confidence:** 3

**Summary:**

The authors present a text classification framework based on measuring similarity between the input documents and multiple related and more specialized concepts that may be associated with label in the label set. To that end, based on my best understanding of the authors proposed method, a LLM is used to generate embedding vectors for each of the concept words (and associated definitions) and the input document. A cosine similarity is calculated between each concept embedding and input document and these similarity scores are then aggregated somehow and passed through an activation function to get final classification scores. There are also some learnable parameters in this setup but reading through the method description, it is unclear how they are actually used (somehow they are used to modify the concept embeddings, I think).

The method is compared to a standard classification fine-tuning setting where the model is trained to predict the classification label using an additional classification head on top of the base LLM. The evaluation is done on 4 sentiment analysis datasets. While the performance improvement is minimal on bert-based models (bert-base and roberta-large), the authors do observe 6-8% gain in absolute terms for t5-encoder.

The authors also perform analysis of what happens when different concept words (or definitions) are used at inference time. They find that definition paraphrasing or using different concept words but belonging to same domain do not impact performance significantly. They also add in some neutral terms to see how model react to those and found that sometimes model can still perform classification with accuracy even when they are not relevant.

**Strengths:**

Originality: The idea of performing text classification by using more description concepts associated with text labels has significant merit. This is especially useful in cases where the labels in label set are vague or overbroad and training set may not cover all possible semantics that may be associated with the given label. As such, a method that allows us to inject more associated concepts at inference time without retraining to capture novel labels / label concept could have significant impact.

**Weaknesses:**

The quality of writing is bad. An informed researcher in this area will have a hard time understanding both methodology and some of the experiments in the paper. In section 3 where authors describe their method, many of the symbols are undefined (for example, n_s, n_y, n_n, n_m, s_D, etc). Similarly, mathematical expressions are written without any context (and it is unclear if they are even meaningnful). For example, expression ∀i∃j • P(C(j), y(i)) has not meaning -- it has quantifiers but doesn't seem to be logical statement). Equations 1-5 that actually describe how embeddings are being used are un-parsable -- I am not sure what any of those expressions means (for example, each expression uses universal quantifier symbol but is not a logical statement). As such, I have no idea what exactly is the method that the authors are proposing (it might be useful to add some pseudocode or something similar in the appendix which might make things clearer).

Moving to evaluation section, in section 5.1, it is unclear how exactly neutral concepts injected (are concepts for positive or negative class replaced with neutral concepts or is it added as a separate class?). Overall, I don't think I can make an educated opinion about the experiment since they depend on understanding the method itself).

**Questions:**

I would like to see a list of definitions from the author about each mathematical symbol introduced in section 3.

---

### Official Review · Reviewer_hACj · 2023-11-09

**Soundness:** 2 fair
**Presentation:** 1 poor
**Contribution:** 1 poor
**Rating:** 3
**Confidence:** 4

**Summary:**

This work introduces a neural sandbox framework for text classification using LLMs by leveraging defined label concepts.
The framework takes a frozen LLM as a vector embedding generator for input text and label concepts and then optimizes an operator to classify the input text based on the relevance scores to the label concepts. The experiments on IMDB, GoEmotion and Amazon Hierarchical Text  Classification show that the framework outperforms three simple classifier baselines.

**Strengths:**

The approach attempts to provide more conceptual explanation, which differs from traditional inherent methods used for explaining text
classification models via attention scores.

**Weaknesses:**

* The major issue to me is the unpair comparsion with the LLM classifier baselines.
For current LLMs, prompts and in-context learning play a key role in their success. Baselines do not have access to the information about label concepts, while the proposed approach has access to them. It is suggested to use those label concepts as at least system prompts along with the input text to feed into the baselines.
* The presentation requires improvement in terms of writing quality and organizational structure.

**Questions:**

Please respond to Weaknesses.

---

### Meta-Review · Area_Chair_A7GT · 2023-12-13

**Metareview:**

The paper presents a text classification framework based on similarity between the input document and a set of related and more specialized concepts that may be associated with a classification label. An LLM is used to embed each concept and the input document, and use the cosine similarity as a score. The experiments consider BERT based models and T5. All reviewers reported key limitations in the experiment set up, paper's presentation and writing, failing to connect with retrieval methods and so on. I agree with all these concerns.

**Justification For Why Not Higher Score:**

NA

**Justification For Why Not Lower Score:**

N/A

---

### Decision · Program_Chairs · 2024-01-16

Reject